# A Systematic Review and Meta-Analysis on the Effect of Nature Exposure Dose on Adults with Mental Illness

**DOI:** 10.3390/bs15020153

**Published:** 2025-01-31

**Authors:** Joanna Ellen Bettmann, Elizabeth Speelman, Annelise Jolley, Tallie Casucci

**Affiliations:** 1University of Utah College of Social Work, 395 South 1500 East, Room 101, Salt Lake City, UT 84112, USA; Joanna.schaefer@socwk.utah.edu; 2Department of Outdoor Education, Georgia College & State University, Campus Box 125, Milledgeville, GA 31061, USA; liz.speelman@gcsu.edu; 3University of Utah J. Willard Marriott Library, 295 S 1500 E, Salt Lake City, UT 84112, USA; tallie.casucci@utah.edu

**Keywords:** nature immersion, nature exposure, mental health, mental illness

## Abstract

Time spent in nature leads to significant physical and mental benefits, but research is mixed on how much time in nature is necessary to affect change in adults’ mental health. This meta-analysis aimed to answer the question: what effect does length and interval of nature dosage have on adults with mental illness? The authors defined nature exposure as an experience in nature lasting at least 10 minutes and taking place in an actual natural setting. Because some studies indicated single experiences of exposure to nature (*one-time*) while others utilized multiple exposures to nature (*interval*), these studies were separated to determine differences between one-time versus interval exposure to nature. Following *Cochrane Handbook for Systematic Reviews of Interventions* and PRISMA reporting guidelines, this review included 78 studies published between 1990 and 2020. The present study found that one-time and interval nature exposure yielded different results for adults with a diagnosed mental illness and adults with symptoms of mental illness. Notably, shorter nature exposure delivered in intervals appeared to show positive significant effects, even more than one-time exposure. This finding has important implications for public health and green space preservation, as being outside for as little as 10 minutes and even in urban nature can improve adults’ mental health.

## 1. Introduction

Existing research indicates that time spent in nature leads to significant physical and mental health benefits ([9]; [25]; [27]; [34]; [42]). Nature exposure appears to benefit human health in a range of ways, including improved cognitive functioning, improved brain functioning, decreased blood pressure, improved physical health, improved sleep, and improved mental health symptoms ([17]). These benefits include decreased tension, anxiety, depression, anger, hostility, fatigue, and confusion; but, research also shows that nature exposure can improve humans’ resiliency against cancer and other illness ([20]; [35]). Even having green and natural spaces in one’s neighborhood appears to have quantifiable mental health benefits ([17]).

### 1.1. Type of Nature Exposure

Some existing research explores the relationship between mental health and type of nature exposure, such as passive or active nature exposure ([14]; [22]; [39]; [41]). There is debate about which type of nature exposure is most beneficial for well-being ([14]; [27]; [41]). Some argue that passive nature exposure is beneficial while others suggest that being in nature while engaging in physical activity shows greatest benefits ([14]; [27]; [41]). One systematic review reported that both being physically active in natural environments as well as simply viewing or sitting in nature had significant effects on participants’ mental health ([39]). The World Health Organization defines mental health as “a state of mental well-being that enables people to cope with the stresses of life, realize their abilities, learn well and work well, and contribute to their community” ([43]).

In a study on the effects of active and passive nature exposure on college students’ health, the authors defined active nature exposure as engaging in physical activity (such as hiking, walking, biking, etc.) in green spaces for more than fifteen minutes per day ([14]). They defined passive nature exposure as engaging in non-physical activities (such as socializing, studying, sitting, etc.) in green spaces for more than fifteen minutes a day. In their study, active nature exposure (compared to passive nature exposure) yielded the greatest benefits for participants’ health and well-being ([14]). Other research considers whether virtual or real nature exposure is optimal. In one study, participants’ positive affect and reflection skills improved following both virtual and real nature exposure ([22]).

### 1.2. Nature Dosage’s Effect

Existing research shows that humans benefit from nature exposure, whether that exposure is ten minutes long or lasts multiple days ([24]; [41]). Studies have examined one-time immersive nature exposures, as well as *interval* (meaning shorter but repeated) nature exposure ([24]; [41]). While research clearly indicates that there are mental health benefits to nature exposure, there is no consensus on the nature dose and dosage frequency with which people should immerse in nature in order to experience benefits.

In one study which surveyed wilderness visitors at a trailhead, researchers found that the length of nature exposure, quantified as either day use or overnight use, showed no relationship to participants’ self-reported stress reduction or mental rejuvenation ([10]). Another study measured salivary cortisol and alpha-amylase, physiological biomarkers of stress, before and after nature exposure ([15]). This study defined nature exposure as spending time outdoors in a place where the participants felt a sense of contact with nature for at least 10 minutes ([15]). The study found that nature exposure decreased salivary cortisol by 21% and salivary amylase by 28%, suggesting significant stress reduction following even brief nature exposure ([15]). The researchers reported even more significant stress reduction with nature exposure between 20 and 30 minutes ([15]).

Research shows benefits from one-time nature immersive experiences, termed *one-time* nature exposure in our study ([3]; [24]; [39]). For example, a systematic review of patients with somatic disease found that patients with high blood pressure and depression benefited from being in nature for at least 30 minutes ([39]). A scoping review of nature exposure for young adults found specific health benefits for different doses of nature and different nature-based activities ([24]). For example, 10 to 30 minutes of sitting outdoors resulted in lower cortisol levels and blood pressure ([24]). In a meta-analysis, [3] ([3]) explored the effect of different doses of exercise in nature (green exercise) on mental health. The authors found that a five-minute episode of green exercise significantly improved participants’ self-esteem and mood. Researchers also noted improvements following longer episodes of green exercise, approximately one-hour-long activities or activities that lasted half the day, but found smaller effects for these activities ([3]). The authors conclude, “there appears to be an immediate effect obtained from the start of green exercise” ([3]), commenting that whole-day activities are quite different and involve different objectives and dynamics.

Some research also examines nature exposure delivered in *intervals*; nature exposure in smaller doses over multiple days. This kind of nature exposure can be measured by adding up the total minutes of daily or weekly nature exposure over a longer time period ([41]). In one study examining *interval* nature exposure, researchers found that, when participants reported at least 120 total minutes of *interval*-delivered nature exposure over a week, they had greater likelihood of reporting good health and well-being ([41]). When participants reported less than that (reporting only one to 119 minutes of recreating outside a week), they were no more likely to report good health or better well-being compared to those reporting zero minutes of time outside ([41]). Notably, this finding was consistent among participants regardless of whether they lived near or far from greenspaces, how much money they made, or if they met the recommended exercise level in the previous week ([41]), suggesting the robustness of the finding. Similarly, [32] ([32]) explored the relationship between the frequency and intensity of nature exposure and health among urban residents. The authors found that participants who spent more total minutes in green spaces reported lower levels of depression and blood pressure. The researchers concluded that participants who spent at least thirty minutes or more in green spaces per week reduced their rates of depression and high blood pressure measurably ([32]).

Researchers define and measure nature dosage and its effects in a range of ways ([3]; [13]; [33]; [41]). [3] ([3]) measured nature dosage in terms of nature exposure duration, intensity of a nature-based activity, and type of green space. [41] ([41]) similarly gauged nature dosage as minutes spent recreating in a natural environment. [32] ([32]) measured nature dose of nature by considering the quality of nature, as well as the duration and frequency of nature exposure.

[32] ([32]) explored the health benefits of nature exposure dosage for individuals living in urban settings. They utilized dose–response analysis, concluding that longer duration nature exposure linked to lower depression prevalence and lower blood pressure, as well as increased physical activity. They noted that nature exposure duration and frequency were both linked to higher levels of physical activity. The authors concluded, “there could be up to 7% fewer cases of depression and 9% fewer cases of high blood pressure if the entire sampled population met the minimum [nature exposure] duration criteria of 30 minutes or more” ([32]). However, their study considered only adults living in Brisbane, Australia, and did not examine adults with mental illness specifically.

In another study examining the effects of nature dosage, White et al. defined nature exposure as minutes spent recreating in a natural environment ([41]). Using cross-sectional data from a national study of British adults, this study concluded that recreating outdoors for at least 120 minutes was most beneficial in improving health outcomes ([41]). Notably, the authors wrote, “Sensitivity analyses using splines to allow duration to be modelled as a continuous variable suggested that beyond 120 min[utes] there were decreasing marginal returns until around 200–300 min[utes] when the relationship flattened or even dropped” ([41]). While valuable, this study examined only adults living in Britain and did not examine adults with mental illness specifically.

These previous studies have considered the effect of nature dosage and noted benefits to mental health. But none appear to have specifically examined the nature dosage effect on adults with mental illness or symptoms of mental illness. This systematic review and meta-analysis aimed to answer the question: what effect does nature dosage length and interval have on adults with mental illness and symptoms of mental illness? With such data, nature exposure interventions could be utilized to help improve the health of this vulnerable population.

## 2. Materials and Methods

The authors conducted their systematic review and meta-analysis with guidance from the *Cochrane Handbook for Systematic Reviews of Interventions for the conduct of our review and meta-analysis* ([12]). For transparency and reproducibility purposes, we adhered to the *Preferred Reporting Items for Systematic Review and Meta-analysis (PRISMA)* reporting guidelines for systematic reviews (PRISMA) and searches (PRISMA-S) in reporting results ([26]; [29]; [31]). The protocol is available at: https://www.crd.york.ac.uk/PROSPERO/display_record.php?RecordID=171549 (accessed on 21 January 2025).

### 2.1. Eligibility Criteria

For the purposes of this review, we defined nature exposure as an experience in nature that lasted a minimum of 10 minutes, without a specified maximum duration, and taking place in a natural setting. This definition is consistent with the definition proposed by [24] ([24]): “green spaces, including manicured urban parks, urban woods, and relatively undisturbed natural sites” (p. 3). Following the precedent set by [2] ([2]), this study included only studies featuring real-life nature exposure, excluding simulated or virtual nature exposure. Following the approach of [40] ([40]), studies that focused on nature connectedness as a personality trait rather than direct exposure to natural environments were excluded. Additionally, studies solely involving indoor plants or passive observation of nature from an indoor environment were omitted.

In order to determine the effect of nature exposure on adults with a range of mental illnesses and symptoms of these illnesses, our study analyzed these two groups separately: adults with symptoms of mental illness and adults with mental illness diagnosed prior to the nature exposure intervention. For the purposes of the present study, we defined mental illness as those mental disorders included in the American Psychiatric Association’s Diagnostic and Statistical Manual 5-Text Revision (DSM-5-TR). In our analyses, we included participants with diagnosed mental illness or symptoms of mental illness, but excluded somatic disorders and developmental disabilities. As in [1] ([1]), we excluded somatic disorders, which are psychiatric disorders manifesting with somatic symptoms, and developmental disorders, which are psychiatric disorders typically first diagnosed in children, in order to focus on the most common DSM-5-TR diagnoses.

The present review encompassed all relevant studies relating to the research questions, which gathered quantitative data regarding nature exposure experiences. These studies used validated assessment tools and included data collection at a minimum of two distinct time points per participant, spanning publishing dates from 1990 to 2020. Studies were excluded that solely collected qualitative data, those conducting data collection at a single time point, and studies pre-dating 1990. The rationale behind excluding pre-1990 works was so that authors could concentrate on recent research, thus making the review’s findings more relatable and applicable to current and future mental health practices. Additionally, studies were omitted if they failed to separate data of adolescent participants from that of adults, did not gather mental health symptom data from participants, failed to differentiate between participants with mental health symptoms and those with diagnosed mental illness, or inadequately defined nature exposure or a specific dose.

### 2.2. Information Sources and Search Strategies

Searches were conducted in the following databases: Medline (Ovid) 1946–2020, Embase (embase.com) 1974–2020, PsycINFO (Ebscohost) 1872–2020, Sociological Abstracts (Proquest) 1952–2020, CINAHL Complete (Ebscohost) 1937–2020, Cochrane Library (wiley.com) 1898–2020, Dissertations & Theses Global (ProQuest) 1861–2020, SportDiscus (Ebscohost) 1800–2020, Scopus (Elsevier) 1970–2020, using a combination of database-specific subject headings and keywords for the concepts of mental illness and nature. Searches were conducted by the fourth author and a research librarian, and were peer-reviewed by an information specialist in accordance with the PRESS guidelines ([23]). Searches were conducted in June 2020. The publication date filter for 1990–2020 was applied. The full search strategies can be found in Appendix A. EndNote x20 (Clarivate) was used for citation management and article duplication removal, with Covidence (Veritas Health Innovation) providing a secondary means for removing duplicates.

### 2.3. Study Selection

The database searches yielded 21,695 results (see Figure 1 for PRISMA flow chart). After duplicates were removed, two reviewers independently screened 14,168 publication titles and abstracts using the systematic reviewing platform Covidence. Lack of consensus was resolved by a third reviewer (the first author). At the next step of the review, two reviewers independently assessed each of 608 full-text publications for inclusion. After full-text assessment, 78 studies were included in the review. Appendix A includes a list of the included studies and their key variables in which the sample had diagnosed mental illness. Appendix A includes a list of the included studies and their key variables in which the sample had symptoms of mental illness. A bibliography of excluded publications at the full-text review stage with reasons for exclusion is included in Appendix A. A bibliography of included studies is included in Appendix A.

### 2.4. Data Review and Extraction Process

Two graduate students manually extracted data from the included studies, including demographic data on participants, outcome data relating to mental health pre and post nature exposure, and data relating to type and length of nature exposure. Adhering to Cochrane recommendations for handling missing data, the authors attempted to contact the studies’ original investigators twice within three weeks to request missing data or information. When authors failed to provide the missing data after these attempts during a three-week period, the article was excluded from the review. Additionally, any studies that lacked means, standard deviations, or sample sizes were excluded from the analysis after reasonable attempts to contact the original authors.

### 2.5. Risk of Bias Assessment

Of the 78 studies meeting the inclusion criteria, 36 were categorized as non-randomized while 43 were categorized as randomized. The authors employed the RoB 2 tool ([37]) to evaluate the risk of bias in the studies with randomized trials and the ROBINS_I tool ([36]) to evaluate the risk of bias in studies with non-randomized trials. For the 36 non-randomized publications, bias was evaluated using the ROBINS-I tool, revealing nine of studies as susceptible to bias. Eight of the studies were at-risk for bias due to missing data. Additionally, two of the six were at-risk for bias related to deviations from intended interventions. Two studies were at-risk for bias due to participant selection, and one of these was also at risk for bias in the classifications of intervention. Regarding the 43 randomized control group publications, assessment was performed using the RoB 2 tool. Eleven out of the 43 randomized control group studies were found to be at-risk for bias. Of these, all eleven were at risk for bias in the randomization process. Additionally, three of these studies were flagged for bias due to deviating from the intended intervention. One study was identified also as having potential bias in outcome measurements.

### 2.6. Syntheses and Analyses

Two unique meta-analyses were conducted based on the extracted pre and post nature exposure data and differentiated by the populations assessed. One analysis measured the change in adults diagnosed with mental illness and the other measured the change in adults with symptoms of mental illness. Effect sizes were calculated from the extracted pre and post nature exposure data to determine the strength of change in the participants. Hedge’s *g* was used to report effect sizes as it is a more conservative form of effect size than Cohen’s *d* ([7]) because it corrects for the wide range of sample sizes found in the included studies.

The data extracted from the original studies used to calculate effect sizes included means and standard deviations at both pre and post testing, the sample size(s), the direction of the effect, and the correlation between the two times or groups. The Comprehensive Meta-Analysis Version (CMA) 4 software ([8]) was used to calculate effect sizes for each study and compute overall meta-analysis statistics. A random-effects model was used as the studies found in these analyses are assumed to be a random sample from a universe of potential studies.

To determine the relationship between the nature dosage for each population and the effect size, meta-regressions were conducted using the CMA 4 software. Regression coefficients were calculated to determine the strength of those relationships. Z-value tests and *p*-values were recorded to assess the significance of these relationships. Additional sub-group analysis was conducted where clear gaps in time was observed between studies. For the purposes of the present study, the authors defined *one-time* nature exposure as spending a discrete amount of time outside at once, while *interval* nature exposure refers to nature exposure over multiple days, totaling a certain amount of time. Because some studies indicated single experiences of exposure to nature (*one-time*) while others indicated multiple exposures to nature (*interval*), these studies were separated to determine if there was a difference between one-time versus interval exposure to nature.

### 2.7. Transparency

The study was approved by a large Western U.S. institutional review board. The quantitative data, analytic methods, and coding that support the findings of this study are available from the first author upon reasonable request. Other studies using different data from this systematic review and meta-analysis have been published separately ([5], [6]; [21]).

## 3. Results

### 3.1. Sample of Studies Included in Meta-Analyses

Overall, there were a total of 78 studies included in this meta-analysis with a total of 4987 participants. The average age of the participants was 39.66 years (SD = 13.72). The total nature dosage in all studies was 230,921.68 minutes; the average nature dosage per study was 2483.03 minutes. The nature dosage in studies examining *one-time* nature exposure ranged from 10 minutes to 10,080 minutes (see Appendix A). The nature dosage in studies examining *interval* nature exposure ranged from 22 minutes to 1120 minutes (see Appendix A).

For the meta-analysis examining participants with diagnosed mental illness, there were 45 studies, coming from 42 articles, which involved 1781 participants. The average age of the participants was 42.07 years (SD = 12.46). The total amount of nature dosage for all of these participants was 206,273.50 minutes, while the average nature dosage was 4583.86 minutes.

For the studies in which participants had a diagnosed mental illness, 40% of the studies focused on participants with mood disorders including depression, affective disorders, bipolar disorder, and seasonal affective disorder. Stressor-related disorders, including PTSD and adjustment disorder, accounted for 8.9% of the studies. In 37.8% of the studies, the sample population either included a range of mental disorders or numerous comorbid mental illnesses. The other 13.3% of the studies included single studies with participants diagnosed with avoidant personality disorder, schizophrenia, ADHD, binge eating disorder, and psychotic disorders.

For the meta-analysis examining participants with symptoms of mental illness, there were 48 studies, coming from 37 articles, which included a total of 3206 participants. The average age of the participants was 37.25 years (SD = 14.62). The total nature dosage was 24,648.18 minutes, while the average nature dosage was 513.50 minutes.

### 3.2. Nature Dosage Effects

Results of the meta-regressions of effect size and nature dosage showed several important effects, as well as some non-significant effects (see Table 1). Of the studies examining participants with diagnosed mental illness, there was no discernable relationship between time and effect size in either a positive or negative direction when all studies with *interval* nature exposure were included. The term *interval* means the nature exposure was delivered over multiple days, weeks, or months (see Figure 2). However, there was a positive and statistically significant relationship between time and effect size from 10 minutes to 600 minutes when nature exposure was *interval* (see Figure 3). This finding indicates that the more time spent in nature, the larger the effect—up to 600 minutes. Notably, there were no studies in this meta-analysis examining *interval* nature exposure with a nature dosage of 600–720 minutes for participants with mental illness. However, studies in this meta-analysis which examined *interval* nature exposure of 720 minutes or more for participants with diagnosed mental illness showed no discernable relationship between time and effect size, either in a positive or negative direction.

When assessed all together, the studies examining participants with diagnosed mental illness with a *one-time* nature exposure showed no discernable effect of nature exposure time on measured outcomes, either in a positive or negative direction (see Figure 4). One study, which included participants who were engaged in nature for so long that the analysis was necessary to be conducted in hours rather than minutes, was considered to be an outlier and therefore was removed from the analysis. Even with the removal of this study from the analysis, there was no significant impact of time in either a positive or a negative direction (see Figure 5). However, there was a positive and statistically significant relationship between time and the effect size from 10 minutes to 105 minutes when the nature exposure was *one-time* (see Figure 6). Notably, the studies in this meta-analysis included multiple studies with nature exposure up to and including 105 minutes, but none between 105 and 960 *one-time* minutes of nature exposure. The studies of participants with diagnosed mental illness which examined *one-time* nature exposure of more than 960 minutes showed no discernable effect of nature exposure time on measured outcomes, either in a positive or negative direction.

Of the studies which examined adults with only symptoms of mental illness, there was a positive significant relationship between time and effect size for all studies with *interval* nature exposure. For these studies, total dosage of the nature exposure (delivered in intervals) ranged from 120 to 1120 minutes. For these studies, more time in nature resulted in larger effect sizes (see Figure 7). Among the studies examining adults with symptoms of mental illness, there was no discernable relationship between time and effect size when the nature exposure was *one-time* even when studies with significantly longer time in nature were removed (see Figure 8 and Figure 9).

## 4. Discussion

This study synthesized existing research to determine the impact of nature dosage on adults with diagnosed mental illness and symptoms of mental illness. Of note, we analyzed data from adults with symptoms of mental illness separately from adults with diagnosed mental illness. We made this decision because, in studies where the adults had symptoms of mental illness, the sample was most often drawn from the community or higher education settings. By contrast, in studies where the adults had diagnosed mental illness, the sample was mostly drawn from healthcare settings including treatment centers. Thus, analyses were intended to capture the effect of nature dosage on two different groups. Results from the present study show that both interval and one-time nature exposure had positive effects on adults with diagnosed mental illness, while only interval nature exposure showed positive effects on adults with symptoms of mental illness. These findings align with previous research indicating that spending time in nature results in significant mental health benefits ([17]; [25]; [42]).

Importantly, our study distinguished between *one-time* and *interval* nature exposure. We found that one-time and interval nature exposure yielded different results for adults with diagnosed mental illness and adults with symptoms of mental illness. This study also found that, for individuals diagnosed with mental illness, exposure to nature lasting ten minutes to two hours had a significant impact. However, it is unclear why longer periods in nature did not have the same significant effect. Notably, this finding from our study was congruent with [41] ([41]), which found in a cross-sectional national sample of British adults that reached 120 minutes per week of outdoor recreation showed the strongest benefit for participants. The authors found “decreasing marginal returns [of nature exposure] until around 200–300 min[utes] when the relationship flattened or even dropped” ([41]). The present study thus is congruent with White et al.’s findings, suggesting that longer nature exposure appears not to show greater benefits.

The overall effects of nature exposure for participants with diagnosed mental illness and symptoms of mental illness are positive and significant. These findings from our study suggest robustly that nature exposure has a significant positive impact on individual mental health.

### 4.1. Adults with Diagnosed Mental Illness

In adults with diagnosed mental illness, our study found that for up to 105 minutes of *one-time* nature exposure, increasing nature exposure time increased positive effect. These findings line up with other research highlighting the benefit of spending a discrete amount of time in nature ([3]; [24]; [39]). However, the present study’s finding that increasing time in nature up to 105 minutes leads to more benefit is in contrast with [3] ([3]) finding that the greatest benefit from nature exposure is in the first five minutes with smaller positive improvements over the next hour of nature exposure and up to a half day of nature exposure ([3]). One meta-analytic study found that participants with mental illness showed the greatest improvement from nature exposure ([3]).

Importantly, our study’s findings differ from [39]’s ([39]) systematic review. [39] ([39]) found that a minimum of 30 minutes of nature exposure was necessary for health benefits in patients with depression and blood pressure ([39]). Though the dosage recommendations vary across these studies, the conclusion that spending a discrete amount of time in nature is beneficial is unanimous ([3]; [24]; [39]). Going outside, even for just 10 minutes, has a positive impact on mental health. This finding supports [24] ([24]) that even 10 to 30 minutes of time outdoors has an impact on college students’ mental health and well-being ([24]).

Our study also found that 960 minutes or more of one-time exposure had neither a positive nor negative effect on adults with diagnosed mental illness. In addition to the finding that individuals derive increasing benefit from nature exposure for up to 105 minutes, this finding indicates that the positive impact of nature exposure does not require copious amounts of time. Rather, benefit is derived from less than an hour of nature exposure. This finding from our study is consistent with research noting substantial benefits from short periods of nature exposure ([3]; [4]; [24]; [39]).

In adults with diagnosed mental illness, our study showed a time-by-dosage effect for interval exposure up to 600 minutes. Increasing the total time outside, up to 600 minutes, improved mental health outcomes. The smallest interval examined was 10 minutes. This finding aligns with [32]’s ([32]) finding that spending an average of 30 minutes outside a week decreased the prevalence of depression in their population ([32]). The present study’s findings furthered the research by clarifying that the benefits of nature exposure are improved with dosage up to 600 minutes of interval exposure. This finding is particularly salient because increasing the frequency or duration of nature exposure has a positive impact on mental health.

### 4.2. Adults with Mental Health Symptoms

In adults with mental health symptoms, our study found a dose effect for *interval* nature exposure from 120 minutes to over 1000 minutes. This finding of our study appears to align with [41]’s ([41]) findings which indicated that participants who spent at least 120 minutes in interval nature exposure evidenced significant health and mental health benefits. In White et al.’s study, participants who engaged in 0 to 119 minutes of *interval* outdoor recreation over the course of a week showed no significant health benefits.

This finding from our study regarding interval nature exposure is important because participants showed benefits from repeated nature exposures. Thus, adults with mental health symptoms are likely to experience some immediate relief from their symptoms if they venture outdoors. They do not need to go deep in the wilderness, mountains, or forest. Rather, they can experience benefits simply from being exposed repeatedly to nearby natural areas. Nature exposure in urban parks may be accessible to those who cannot access or do not want to spend time in wilderness areas.

Notably in our study, *one-time* or one-time nature exposure did not show a significant effect on individuals with symptoms of mental illness. This finding does not align with existing research showing a benefit of spending a discrete amount of time in nature ([3]; [24]; [39]). Notably, many of the studies included in our meta-analysis utilized quite brief nature exposure. This finding regarding *one-time* nature exposure highlights the importance of repeated, or *interval*, nature exposure for adults with mental illness. When adults with mental illness are exposed to nature repeatedly over time, they appear to benefit significantly.

### 4.3. Healthcare Implications of the Present Study

Findings from our study suggest the importance of getting outdoors to improve mental health among adults with mental illness or symptoms of mental illness. One way to encourage adults to get outdoors is to utilize nature prescription programs. These programs promote nature exposure as a health intervention by encouraging healthcare providers to prescribe nature exposure to patients ([16]; [19]; [24]). These programs, coupled with the findings from our study, equip healthcare providers with the rationale and the means to prescribe nature as a healthcare intervention. Healthcare providers should be aware that only a short time in nature can significantly improve symptoms for individuals with mental illness. Thus, prescribing nature as a health intervention emerges as an accessible and effective adjunct to traditional treatment.

### 4.4. Public Health Implications of the Present Study

Findings from our study also have important implications for public health. This meta-analysis incorporated studies involving as little as 10 minutes of nature exposure, including urban natural areas. However, economically-disadvantaged groups face challenges in accessing green spaces ([18]). These challenges include limited availability of green spaces within many communities ([18]). Notably, a systemic review suggests that people in lower socioeconomic communities experience greater health benefits from public green spaces compared to individuals in more affluent communities ([30]). One study found that if all respondents lived in neighborhoods with 20% vegetation coverage, then the total number of respondents showing symptoms of depression likely would decrease by 11% ([11]).

The economic projection of mental health service costs in the US was $280 billion for 2020 ([38]). This massive cost of mental health services may increase as mental illness prevalence is increasing ([38]). Prescribing nature exposure as an adjunct to treatment for those with mental illness potentially could ameliorate this rising cost ([28]). Research showed a 28% reduction in demand for general practitioner services after a social prescription program, such as nature prescription ([28]).

## 5. Limitations of the Present Study

The present study considered research on the immediate effects of nature exposure and so was unable to answer whether nature exposure affects lasting impact on adults with mental illness. Additionally, most studies in this meta-analysis did not report the racial or ethnic make-up of their sample, nor the income level, sexual orientation, or gender identity of their participants. Thus, our study was unable to examine the differential effect of nature exposure’s effect on a range of populations. Future studies should examine this important point.

## 6. Future Research

Future research should define nature exposure and its specific benefits more clearly for a range of populations. Specifically, future research should explore nature’s effect on adults with mental illness who have a range of identities and sociocultural differences in order to ensure that healthcare providers are utilizing this potential intervention to maximal benefit for a range of populations. Additionally, future research should explore how individuals of different sexes and gender identities respond differently to nature exposure in order to understand best who benefits from these interventions. For example, nature prescription programs may be most useful or well-received in certain populations, but research has not explored this area fully.

## 7. Conclusions

Spending time in nature has significant physical and mental benefits for adults, but the amount of time needed for positive effects on those with mental illness has been unclear until now. Our study found significant effects of nature exposure for adults with mental illness or symptoms of mental illness, suggesting the importance of utilizing nature prescription within healthcare settings.

## Figures and Tables

**Figure 1 behavsci-15-00153-f001:**
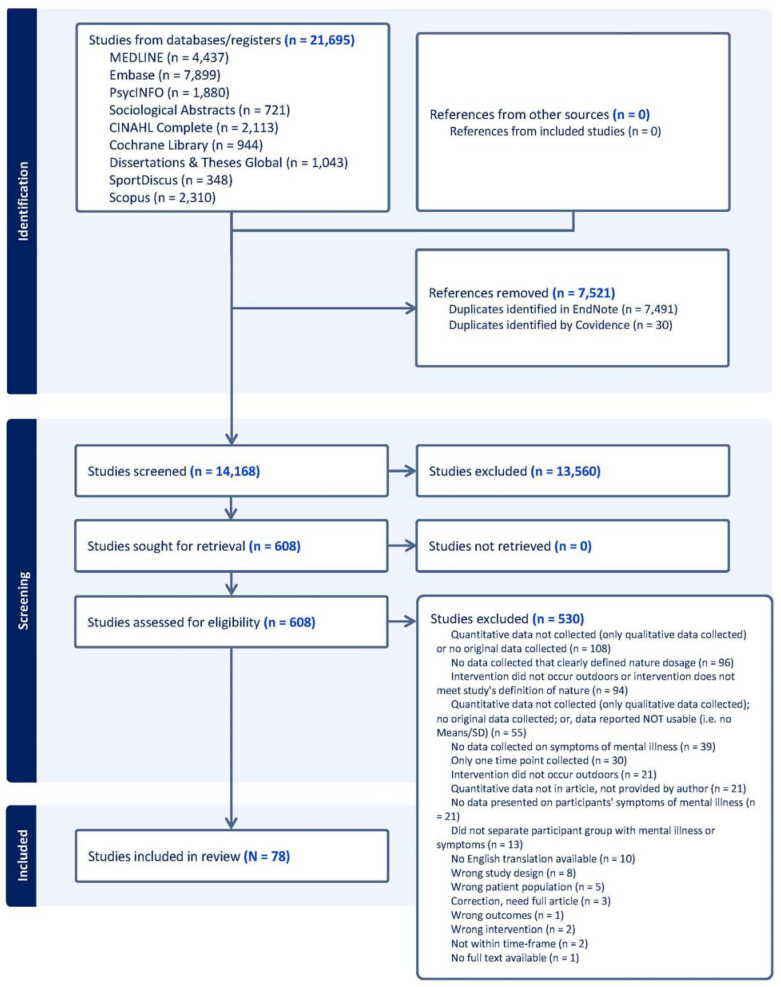
PRIMSA flow diagram.

**Figure 2 behavsci-15-00153-f002:**
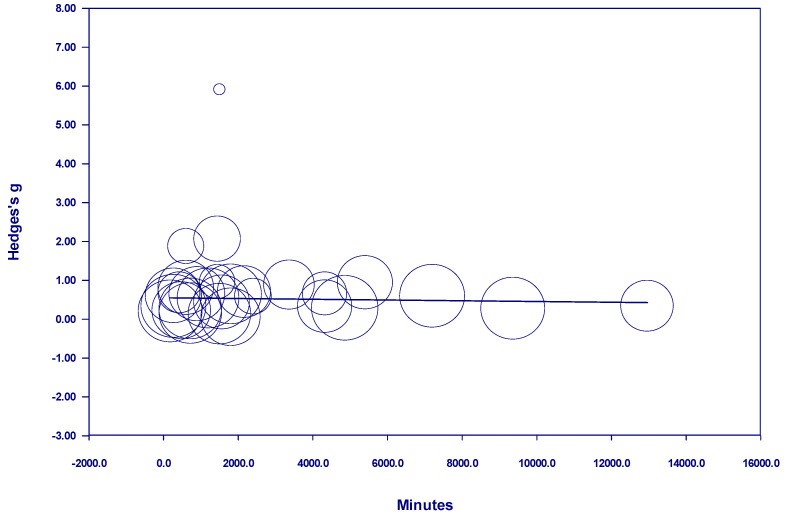
Regression of Hedges g on interval nature dosage in studies of participants with diagnosed mental illness.

**Figure 3 behavsci-15-00153-f003:**
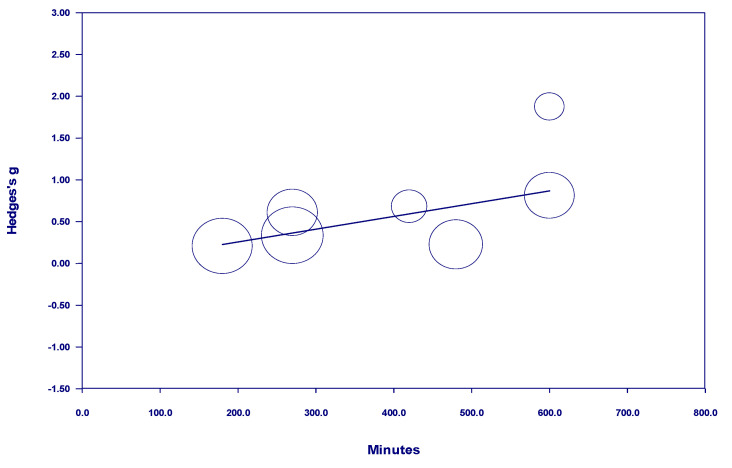
Regression of Hedges g on interval nature dosage (of 600 minutes or less) in studies of participants with diagnosed mental illness.

**Figure 4 behavsci-15-00153-f004:**
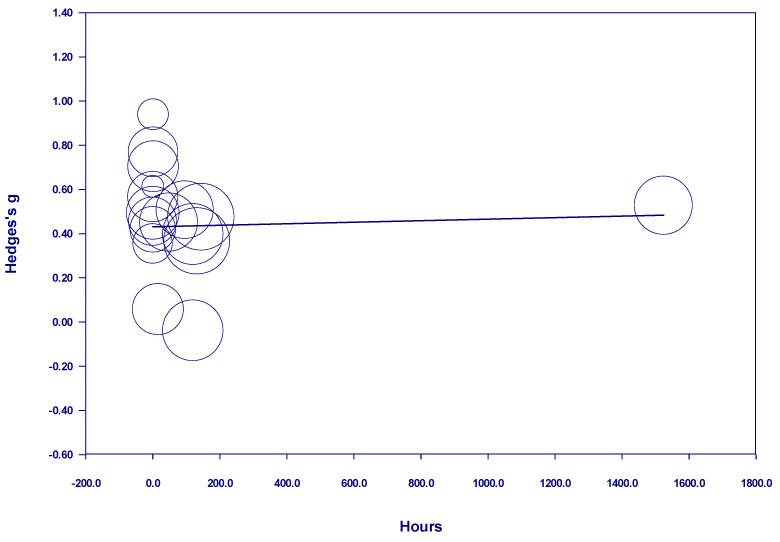
Meta-regression of Hedges g on one-time nature dosage hours in studies of participants with diagnosed mental illness.

**Figure 5 behavsci-15-00153-f005:**
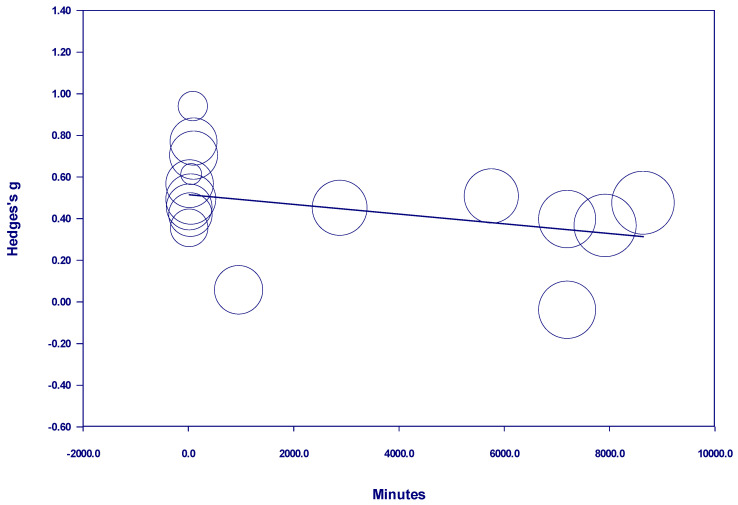
Regression of Hedges g on one-time nature dosage minutes in studies of participants with diagnosed mental illness (without outlier).

**Figure 6 behavsci-15-00153-f006:**
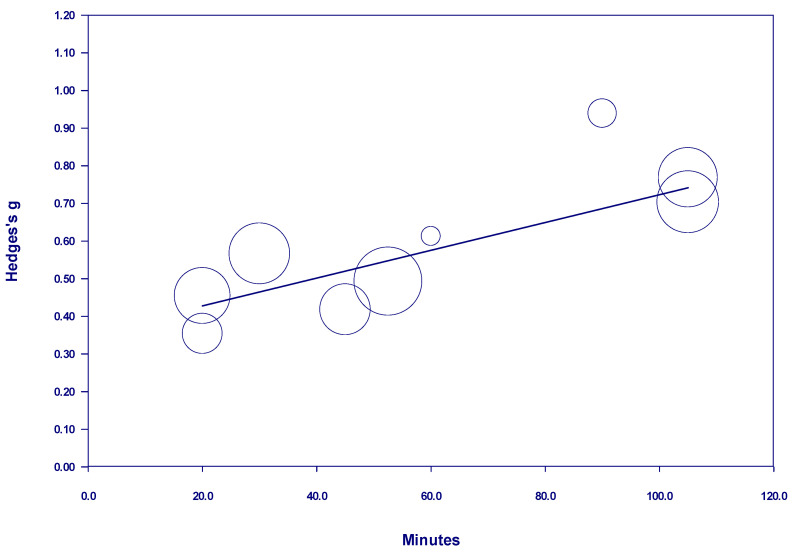
Regression of Hedges g on one-time minutes of activity in nature (less than 120 minutes) in studies of participants with diagnosed mental illness.

**Figure 7 behavsci-15-00153-f007:**
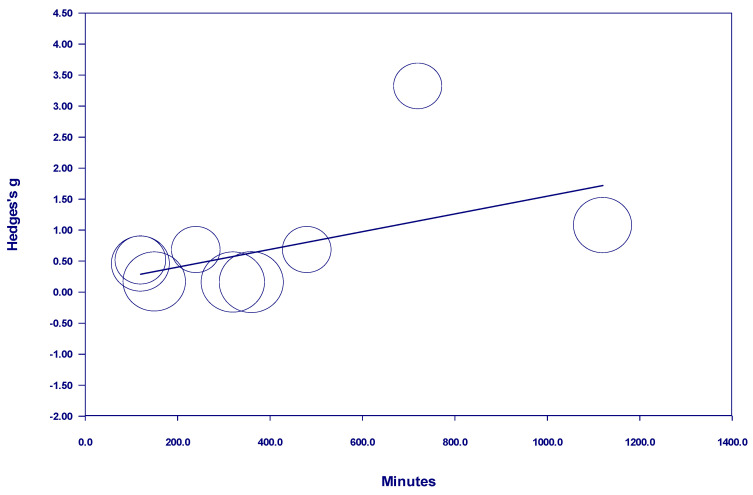
Regression of Hedges g on interval nature exposure in studies of participants with symptoms of mental illness.

**Figure 8 behavsci-15-00153-f008:**
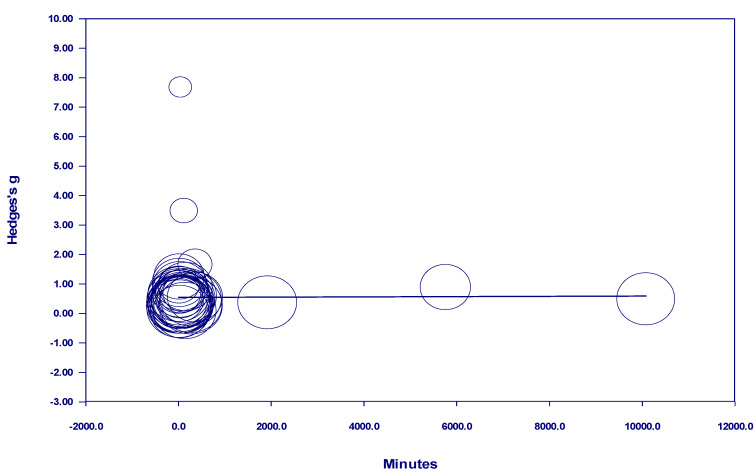
Regression of Hedges g on one-time nature exposure total time in studies of participants with symptoms of mental illness.

**Figure 9 behavsci-15-00153-f009:**
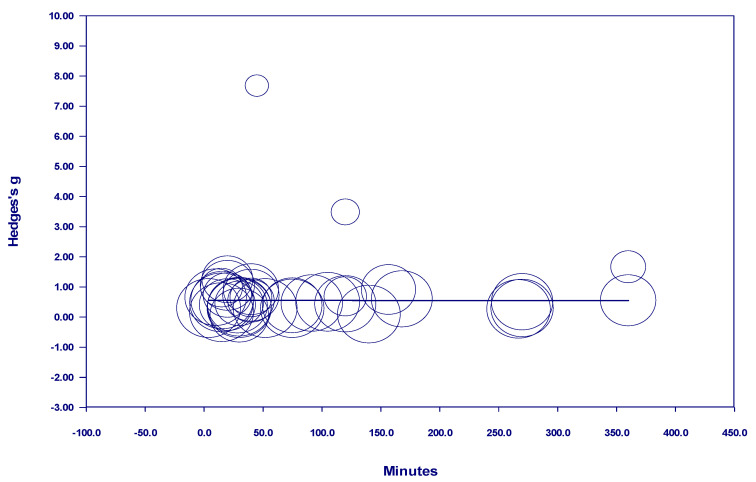
Regression of Hedges g on one-time nature exposure total time in studies of participants with symptoms of mental illness (without the longest dosage studies included).

**Table 1 behavsci-15-00153-t001:** Co-efficient and Z-value of interval and one-time nature exposure.

	Co-Efficient	Z-Value (*p*-Value)
**One-time nature exposure**		
Continuous hours of nature exposure in studies of participants with diagnosed mental illness	0.0000	0.26 (0.7962)
Continuous minutes of nature exposure in studies of participants with diagnosed mental illness (without outlier)	−0.0000	−1.54 (0.1234)
Continuous minutes of nature exposure in studies of participants with diagnosed mental illness (less than 2 h)	0.0037	2.10 (0.0358) *
Continuous minutes of nature exposure in participants with symptoms of mental illness	0.0000	0.16 (0.8704)
Continuous minutes of nature exposure (without the longest studies) in participants with symptoms of mental illness	−0.0000	−0.01 (0.9904)
**Interval nature exposure**		
Interval nature exposure (in minutes) in all studies of participants with diagnosed mental illness	−0.0000	−0.47 (0.6362)
Interval nature exposure (in minutes) of less than 600 minutes in studies of participants with diagnosed mental illness	0.0015	2.19 (0.0287) *
All studies of interval nature exposure (in minutes) with participants who have symptoms of mental illness	0.0014	3.04 (0.0024) *

* significant at 0.05.

## Data Availability

No new data were created or analyzed in this study.

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
