# Peer review of "A Systematic Review and Meta-Analysis on the Effect of Nature Exposure Dose on Adults with Mental Illness"

_behavsci, 2025, doi:10.3390/bs15020153_

Round 1

Reviewer 1 Report

Comments and Suggestions for Authors

Dear Authors, 

Thank you for the opportunity to review this meta-analysis. This is an important research area, and a meta-analysis exploring this topic is welcomed. The methods of the systematic review and reporting of results is thorough. I have provided a number of comments below to further improve the impactfulness of this review. Majority of these relate to rewording sections of the introduction, as some of the descriptions of prior research is confusing and clearer definitions of key concepts need to be included. I hope you find these comments useful and I wish you the best in further improving this paper.

Line(s)

Comment

38-39

It would be useful to quantify what ‘being near nature’ means in this sentence.

47-56

As it is explained what Holt et al define active vs passive nature exposure was, it would be useful to explain how the systematic review Trostrup et al. defined these two concepts too.

62-63

The definition of interval being “meaning weekly or occasional” is confusing. Could interval also be daily for example? Need a clearer definition here.

78-79

The term continuous nature exposure also needs to be more clearly defined. As this is a one-time experience, the term continuous seems inappropriate. Could it be referred to as adhoc or something similar?

84-109

These discussions of research need to be reworded to improve clarity. For example did the Barton & Pretty 2010 study find that 5 minutes had significant improvements, but one hour + nature exposure had less improvements? This is how this information reads, and if that is the case that should be unpacked more as is an interesting finding.

Also did the White et al., 2019 student find that people with 0-119 minutes of nature exposure a week had no improvements, but 120 minutes did have improvements? Again if this is not the case it should be reworded

110-137

Consider combining the information under “continuous vs interval nature exposure” and “Nature dosage’s Effect” into one section as the distinction between these concepts is unclear and therefore confusing to be under two different headings. There also needs to be a explicit definition of what dosage is in this context.

139-144

Given that this review focuses on mental illness, there needs to be a clear definition of mental illness in the introduction, and importantly a discussing regarding the difference between mental health and mental illness.

167

Please provide an explanation / example of what somatic disorders and developmental disabilities are- and justify why these were not included.  

Figure 1

The arrow from identification box to the references from other sources box should be edited to be consistent to the other arrows

Also the n in “studies included in review (n = 79) should be a capital N.

329

Should also include SD for age of participants.

347

The definition of interval should be put into the syntheses and analyses section rather than the results section. A definition of continuous should also be put into the syntheses and analyses section.

326-341

It would be useful to include additional information into the sample of studies included section- such as with diagnosed mental illness were reviewed in the research?

521-524

This sentence, “The present study found that, for individuals diagnosed with mental illness, exposure to nature for less than two hours had a significant impact.” Needs to be reworded for clarity- e.g. does this suggest that people who had nature exposure less than two hours saw positive impacts?

Discussion

There should be a couple of sentences in the discussion that compare the results for mental illness vs mental illness symptoms- tell the readers why these were viewed separately and what do the results tell us?

631

It is stated that most of the studies did not report gender- but in the supplementary materials the vast majority of studies did report the gender of their participants. Therefore gender differences could be explored.

Author Response

To the Editor and reviewers:

Thank you very much for your careful review of our work. We appreciate the opportunity to revise our work. We now have revised the manuscript in accordance with reviewer feedback, using “track changes” in order to indicate our revisions. Below we list each piece of reviewer feedback and how we responded to it with revisions in the manuscript:

Reviewer #1

Line(s)

Comment

Our response

38-39

It would be useful to quantify what ‘being near nature’ means in this sentence.

We revised that sentence in order to increase clarity

47-56

As it is explained what Holt et al define active vs passive nature exposure was, it would be useful to explain how the systematic review Trostrup et al. defined these two concepts too.

Trøstrup et al. (2019) did not use the same terms, e.g. passive versus active nature exposure. So we revised that sentence on Trøstrup et al. to describe how that study classified nature exposure

62-63

The definition of interval being “meaning weekly or occasional” is confusing. Could interval also be daily for example? Need a clearer definition here.

We modified the phrase to read instead “shorter but repeated” in order to increase clarity

78-79

The term continuous nature exposure also needs to be more clearly defined. As this is a one-time experience, the term continuous seems inappropriate. Could it be referred to as adhoc or something similar?

We changed the term “continuous” to “one-time” nature exposure to increase clarity as suggested by the reviewer

84-109

These discussions of research need to be reworded to improve clarity. For example did the Barton & Pretty 2010 study find that 5 minutes had significant improvements, but one hour + nature exposure had less improvements? This is how this information reads, and if that is the case that should be unpacked more as is an interesting finding.

Also did the White et al., 2019 student find that people with 0-119 minutes of nature exposure a week had no improvements, but 120 minutes did have improvements? Again if this is not the case it should be reworded

We added sentences to our discussion of Barton & Pretty in order to increase clarity. The reviewer’s summary of White et al’s findings is correct, so we did not make changes to that

110-137

Consider combining the information under “continuous vs interval nature exposure” and “Nature dosage’s Effect” into one section as the distinction between these concepts is unclear and therefore confusing to be under two different headings. There also needs to be a explicit definition of what dosage is in this context.

We combined these two sections of the literature review, as suggested by the reviewer. The definition of nature dosage is now in lines 114-119.

139-144

Given that this review focuses on mental illness, there needs to be a clear definition of mental illness in the introduction, and importantly a discussing regarding the difference between mental health and mental illness.

We added sentences to the Eligibility subsection of Materials and Methods section in order to more clearly define mental illness. We also added a definition of mental health to the introduction.

167

Please provide an explanation / example of what somatic disorders and developmental disabilities are- and justify why these were not included.  

We added this information to this section, as suggested by the reviewer.

Figure 1

The arrow from identification box to the references from other sources box should be edited to be consistent to the other arrows

Also the n in “studies included in review (n = 79) should be a capital N.

We fixed these two issues as suggested by the reviewer.

329

Should also include SD for age of participants.

We added this to the beginning of the Results section.

347

The definition of interval should be put into the syntheses and analyses section rather than the results section. A definition of continuous should also be put into the syntheses and analyses section.

The definitions of interval nature exposure and one-time nature exposure are in the Syntheses and Analyses subsection of Materials and Methods (not in the Results section).

326-341

It would be useful to include additional information into the sample of studies included section- such as with diagnosed mental illness were reviewed in the research?

We added this information to the beginning of the Results section.

521-524

This sentence, “The present study found that, for individuals diagnosed with mental illness, exposure to nature for less than two hours had a significant impact.” Needs to be reworded for clarity- e.g. does this suggest that people who had nature exposure less than two hours saw positive impacts?

We modified this sentence in order to increase clarity, as suggested by the reviewer.

Discussion

There should be a couple of sentences in the discussion that compare the results for mental illness vs mental illness symptoms- tell the readers why these were viewed separately and what do the results tell us?

We included rationale for the decision to analyze these two groups separately in the first paragraph of the Discussion section. We also included in that first paragraph a summary of the key findings

631

It is stated that most of the studies did not report gender- but in the supplementary materials the vast majority of studies did report the gender of their participants. Therefore gender differences could be explored.

Because this point is beyond the scope of the research question for this study, we added this point to the Future Research section of discussion.

Reviewer #2

  • Introduction: informative and well done. Our response: thank you very much.
  • Materials and Methods: clear, well done. Our response: thank you very much.
  • Results: What type of mental illness was examined? What type of mental disorder was being investigated? Nothing is written about this by the authors. This is very important given that the term ‘mental illness’ covers the entire manual i.e. all disorders according to the DSM-V. We know nothing about the types of mental illness that were considered in the study. What kind of illness do the authors have in mind? How should nature exposure to be differentiated according to the type of disorder? For which types of mental illnesses this nature exposure is advisable and for which it is not. This aspect was completely ignored in the meta-analysis. Our response: we added several sentences to the beginning of the Results section to address the reviewer’s concern. Please note that separate analyses concerning which how mental illnesses responded differently to nature exposure was published separately (Bettmann et al., 2024b) and is beyond the scope of the research question for this study.
  • Authors tend to repeat the phrase ‘the present study’ many times. Any other wording should be used to avoid repeating the same phrase. Our response: we revised the Discussion section to reduce significantly the repetition of that phrase.

Reviewer 2 Report

Comments and Suggestions for Authors

Introduction: informative and well done.

Materials and Methods: clear, well done.

Results:  What type of mental illness was examined? What type of mental disorder was being investigated? Nothing is written about this by the authors. This is very important given that the term ‘mental illness’ covers the entire manual i.e. all disorders according to the DSM-V. We know nothing about the types of mental illness that were considered in the study. What kind of illness do the authors have in mind? How should nature exposure to be differentiated according to the type of disorder? For which types of mental illnesses this nature exposure is advisable and for which it is not. This aspect was completely ignored in the meta-analysis.

Discussion. Authors tend to repeat the phrase ‘the present study’ many times. Any other wording should be used to avoid repeating the same phrase.

Author Response

(The authors gave the same response as above.)
